# Chloroquine decreases cardiac fibrosis and improves cardiac function in a mouse model of Duchenne muscular dystrophy

**Takuya Hirata, Shiro Baba[ID]\*, Kentaro Akagi, Koichi Matsuda, Katsutsugu Umeda, Souichi Adachi, Toshio Heike, Junko Takita**

Department of Pediatrics, Graduate School of Medicine, Kyoto University, Shogoin, Sakyo-ku, Kyoto City, Japan

\* shibaba@kuhp.kyoto-u.ac.jp

**Data Availability Statement:** All relevant data are within the paper and its Supporting Information files.

## Abstract

### Background

Duchenne muscular dystrophy (DMD), a severe degenerative skeletal and cardiac muscle disease, has a poor prognosis, and no curative treatments are available. Because decreased autophagy has been reported to contribute to skeletal muscle degeneration, therapies targeting autophagy are expected to improve skeletal muscle hypofunction. However, the role of this regulatory mechanism has not been evaluated clearly in DMD cardiomyocytes.

### Methods

In this present study, we evaluated myocardial fibrosis and its mechanism in *mdx* mice, a model of DMD, and also evaluated changes in cardiac function.

### Results

As assessed by LC3 immunohistochemistry, a small number of autophagosomes were detected in cardiomyocytes of both *mdx* mice and control wild-type (WT) mice. The number of autophagosomes was significantly enhanced by 4 weeks of isoproterenol-induced cardiac stress in cardiomyocytes of *mdx* but not WT mice. Simultaneously, isoproterenol increased cardiomyocyte fibrosis in *mdx* but not WT mice. Administration of chloroquine significantly decreased cardiomyocyte fibrosis in *mdx* mice, even after isoproterenol treatment. Left ventricle size and function were evaluated by echocardiography. Left ventricular contraction was decreased in *mdx* mice after isoproterenol treatment compared with control mice, which was alleviated by chloroquine administration.

### Conclusions

Heart failure in DMD patients is possibly treated with chloroquine, and the mechanism probably involves chloroquine's anti-inflammatory effects.

**Funding:** This study was supported by Kiban C Kakenhi, supported by Grant-in-Aid Scientific Research (No. P24791059), Sanofi Pasteur Japan (No. GDC160982), and Novartis Pharma Japan. The funder had no role in study design, data collection and analysis, decision to publish for the manuscript.

**Competing interests:** The authors have declared that no competing interests exist.

**Abbreviations:** DMD, Duchenne muscular dystrophy; WT, wild type; iso, isoproterenol; ch, chloroquine; PFA, paraformaldehyde; LV, left ventricle; LVEDd, left ventricular end-diastolic diameter; LVESd, left ventricular end-systolic diameter; IVSd, end-diastolic interventricular septal thickness; LVPWd, left ventricular end-diastolic posterior wall thickness; LVFS, left ventricular fractional shortening; LVEF, left ventricular ejection fraction.

## Introduction

Duchenne muscular dystrophy (DMD) is the most common and severe form of muscular dystrophy, and is caused by mutations in the gene encoding dystrophin located on chromosome Xp21 [1–3]. DMD is inherited in an autosomal recessive manner and is relatively common, with an incidence of approximately 1 per 3,500 male births [4,5].

Muscle degeneration and subsequent fibrosis occur at early ages in DMD patients. Muscle weakness results in walking difficulties, and ultimately in the development of respiratory muscle failure and heart failure. Respiratory and heart failure are common lethal complications of DMD, and frequently affect patients in their late teens and early twenties [1]. Recently, ventilator support devices, such as home ventilators, have been developed, prolonging the mean lifespan of ventilated DMD patients to over 35 years [6–11]. Therefore, cardiomyopathy is now the leading cause of death in DMD patients. The percentage of DMD patients who died from cardiac complications increased from 8% to 44% after the development of home ventilation devices in the 1990s [12].

Fibrosis-associated cardiomyopathy generally leads to dilated cardiomyopathy. More than 80% of DMD patients older than 18 years have reduced cardiac function, and 90% of DMD patients develop dilated cardiomyopathy [13,14]. General therapeutic protocols for dilated cardiomyopathy secondary to DMD are not curative, e.g. combinations of diuretics, vasodilators, and beta-blockers [15,16]. To develop more targeted approaches to treatment, it is crucial to delineate the regulatory mechanisms of cardiomyopathy in DMD. A recent report suggested that cardiomyocyte apoptosis contributes to DMD-induced cardiomyopathy in an *in vitro* study using induced pluripotent stem cells (iPSCs) [17]. In the present study, we focused on another type of cell death, autophagy, and fibrosis in cardiomyocytes to determine if this regulatory mechanism contributed to DMD-induced cardiomyopathy using the *mdx* mouse model of DMD.

## Materials and methods

### Experimental animals

C57BL/6 wild-type (WT) mice were purchased from CLEA Japan, Inc. (Japan). *Mdx* mice, a model of DMD, were bred at Kyoto University. All animal procedures were conducted following the guidelines of the Kyoto University Animal Committee and with prior approval from the Institutional Ethical Committee and an ARRIVE guideline. Animals were handled under the Declaration of Helsinki. When sacrificing mice for experiments, a high-concentration $CO_2$ container was used. All mouse experiments were performed using anesthesia to alleviate pain.

To increase cardiac load, capsulated isoproterenol was inserted subcutaneously from 12 to 16 weeks in both WT and *mdx* mice for a dosage of 0.5 mg/kg/day. For chloroquine treatment, chloroquine was dissolved at 75 mg in 250 mL of clean water and given to mice using an automatic water dispenser. Water loss was checked every other day, and it was the same in all mouse groups. Clinical observations to monitor the side effects, activity, appetite (weight loss), and diarrhea, of all experiments were recorded every other day. Echocardiography was performed at 16 weeks of age, as was histological evaluation. A summarized animal protocol is shown in Fig 1.

### Electron microscopy

For transmission electron microscopy analysis, left ventricular sections from 16-week non-treated WT and *mdx* mice were dissected after sacrifice and fixed for 1 h in a pH 7.4 solution containing 4% paraformaldehyde (PFA) and 0.5% glutaraldehyde in 0.1 M cacodylate buffer.

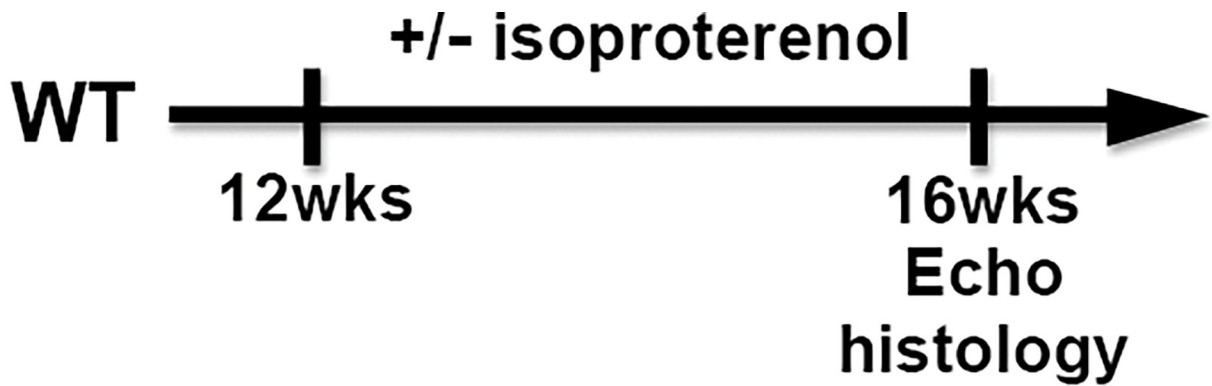

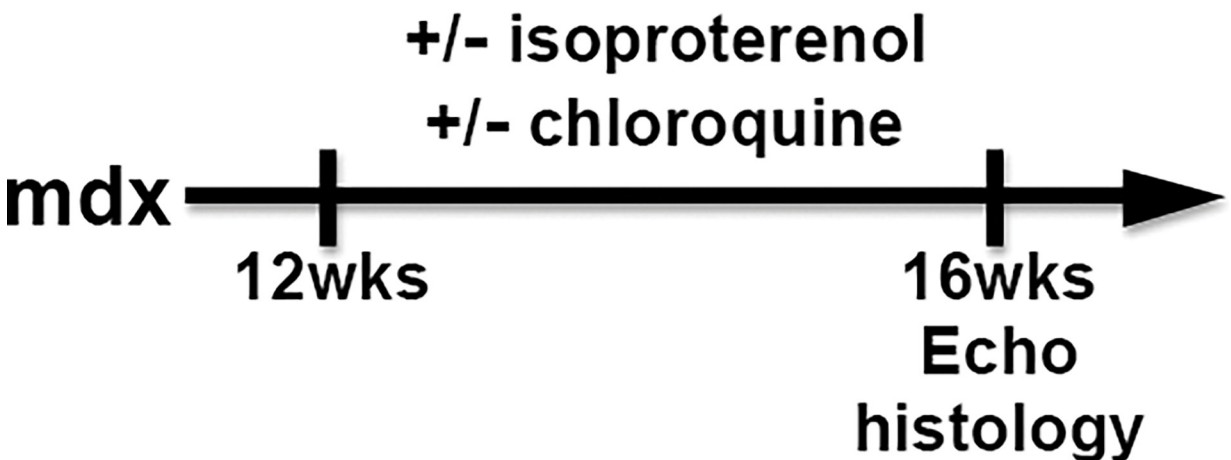

**Fig 1. Animal experiment protocol.**

Ultra-thin sections were cut at a thickness of 7 μm and embedded on glass slides for observation with an H-7650 electron microscope (Hitachi, Ltd., Japan).

### Histology and immunohistochemistry

Mouse hearts were isolated, and 4%-PFA fixed-left ventricles were sliced at 7 μm and stained using a Hematoxylin-Eosin Stain Kit and Picro-Sirius Red Stain Kit (COSMO BIO, Japan). Fibrosis was imaged using a BZ-X-710 microscope (KEYENCE, Japan), and fibrotic areas in each cross-section of the left ventricle at the papillary muscle level were quantified using the automated calculation software of the BZ-X-Analyzer software (KEYENCE, Japan). Autophagosome was stained with LC3 (Santa Cruz, sc-28266) and automatically counted the number of LC3-positive dots by using a BZ-X-Analyzer software and Photoshop software (Adobe).

### Western blotting

Protein lysate was extracted from mouse hearts by using RIPA buffer (Nakalai Tesque, 16488–34) containing 1×protein inhibitor cocktail (1:100; Nakalai Tesque, 25955–24), followed by incubation on ice. Each 30μg protein was loaded onto 12.5% SDS-PAGE gels. The blots were incubated with anti-LC3 antibody (1:1000; Sigma-Aldrich, L8918), anti-p62/SQSTM-1

(PM045, 1:1000; MBL), anti-caspase 3 (1:1000; Cell Signaling, 9662) and peroxidase-conjugated anti-GAPDH monoclonal antibody (1:5000; Fuji Film, 5A12) at 4°C overnight followed by blocking with 5% skim milk. For secondary antibodies, Peroxidase AffinniPure goat anti-rabbit IgG (1:5000; Jackson ImmunnoResearch, 111-035-003) was used and incubated for one hour at room temperature. The membrane was developed using an ECL western blotting substrate (BIO-RAD, 1705061). Blots were imaged using the ChemiDoc XRS+ imaging system (BIO-RAD). For quantification, the signal intensity was quantified using BIO-RAD Image Lab 6.0 software (BIO-RAD), the ratio with GAPDH was determined, and the expression of the ratio of p62/SQSTM-1/GAPDH, the ratio of LC3-II/LC3-I and the ratio of caspase 3/GAPDH were calculated.

## Echocardiography

Hemodynamics were indirectly measured by an echocardiogram with a 50 MHz transducer (Vevo2100; Primetech, UK). Left ventricular end-diastolic and end-systolic diameter (LVEDd, LVESd), end-diastolic interventricular septal thickness (IVSd), left ventricular end-diastolic posterior wall thickness (LVPWd), left ventricular fractional shortening (LVFS), and LV ejection fraction (LVEF) were measured to evaluate cardiac function in 16-week-old mice. During echocardiography, mice were sedated with 1–3% sevoflurane (Maruishi Seiyaku, Japan), with an approximate heart rate of 400 beats per minute.

## Statistics

All experiments used at least 3 mice in each group and were performed 3 or more times. In detail, electron microscopy evaluation, LC3 staining, and Western blotting were performed in three individual experiments from three different mice hearts. For the evaluation of fibrosis of the left ventricle of mice, we conducted experiments with five mice in each group. Regarding the echocardiogram study, experiments were conducted with five mice in each group. Statistical significance was evaluated with a one-way ANOVA followed by the Tukey–Kramer test using JMP® Pro156 (11.0.0) software. $p$-values <0.05 were considered statistically significant.

## Results

### Increased autophagosomes after cardiac stress in *mdx* mouse cardiomyocytes

To determine if autophagy occurred in DMD cardiomyocytes, we first observed autophagosomes in cardiomyocytes of non-treated 16-week-old WT mice and *mdx* mice using an electron microscope (Fig 2A). LC3 positive dots were detected in cardiomyocytes of both WT and *mdx* mice, but the numbers were small and at similar levels (Fig 2A (a) and (c)). After enhancing cardiac stress by administration of isoproterenol for 1 month, the number of LC3 positive dots increased in cardiomyocytes of both 16-week WT and *mdx* mice (Fig 2B (b) and (d)). However, the upregulation ratio of LC3 positive dots was dramatically enhanced by isoproterenol in cardiomyocytes of *mdx* mice compared with those of WT mice. Subsequently, chloroquine, an autophagy inhibitor as well as an anti-inflammatory and anti-immunosuppressive agent, was administered to *mdx* mice simultaneously with isoproterenol for 1 month to verify that the LC3-positive dots were autophagosomes. Administration of chloroquine diminished nearly all LC3 positive dots in 16-week *mdx* mouse cardiomyocytes (Fig 2B (e)). Quantitative evaluation of LC3 dots showed that the number of dots increased significantly in *mdx* mice cardiomyocytes after isoproterenol stress and decreased with chloroquine administration (Fig 2C). As previous papers mentioned that autophagy, assessed by LC3 positive dots, was

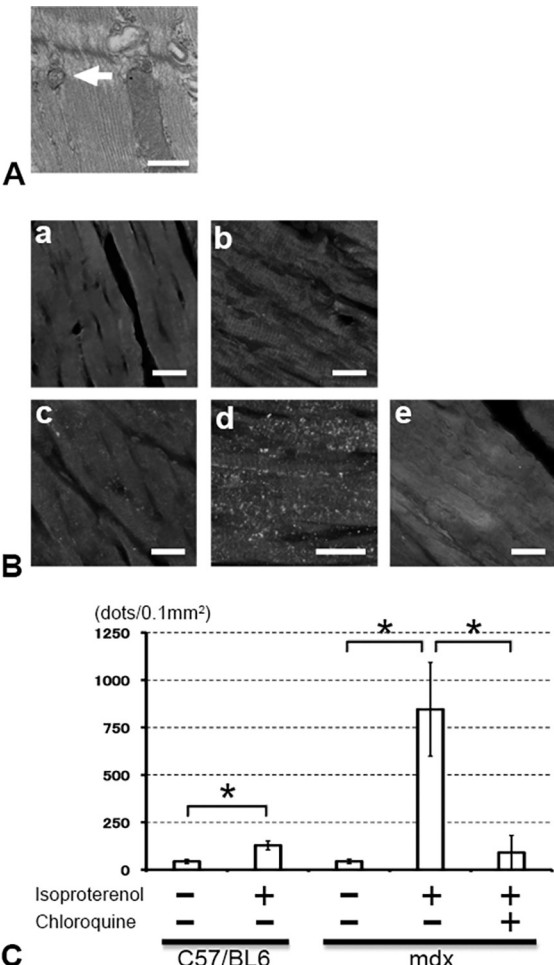

**Fig 2. Autophagosome detection in left ventricle (LV) sections, and LC3 expression in mdx mice cardiomyocytes.**
(**A**) A representative autophagosome in an *mdx* LV cardiomyocyte administrated isoproterenol. (Arrow), Scale bar, 100 nm. (**B**) LC3 positive autophagosomes in LV sections with and without isoproterenol treatment. a) WT untreated, b) WT isoproterenol-treated, c) *mdx* untreated, d) *mdx* isoproterenol-treated, e) *mdx* isoproterenol + chloroquine-treated. Scale bar, 50 μm. (**C**) Number of LC3 positive dots per 0.1mm2 in each LV section. *: $p < 0.05$.

over-activated in *mdx* mice [18], we performed western blotting assessing LC3-I and LC3-II to confirm the LC3 dots results. LC3-I expression was strong and LC3-II expression was weak in WT mice. The expression level ratio did not change even after the administration of isoproterenol with/without chloroquine. When LC3 expression was also evaluated in *mdx* mice, LC3-I expression was stronger than LC3-II expression under non-cardiac stress. After administration of isoproterenol, the expression of LC3-I and LC3-II became almost the same levels. In other words, cardiac stress reduced the expression level of LC3-I and increased the expression level of LC3-II. When an autophagy inhibitor, chloroquine, was administered to *mdx* mice, the ratio of LC3-I and LC3-II expression levels did not change even after the administration of iso-proterenol. We also evaluated the expression of LC3-I and LC3-II in 30-week-old mice, and the expression pattern was similar to that of WT mice (Fig 3A). Cardiomyocyte expression of P62/SQSTM-1 did not significantly change in WT mice after administration of isoproterenol or chloroquine(Fig 3A). The expression levels of LC3-I, LC3-II, and p62/SQSTM-1 were quantified. When the LC3-II/LC3-I ratio and the expression level of p62/SQSTM-1 in the cardio-myocytes of untreated WT mice were set as 1, and the ratios of other expression levels were

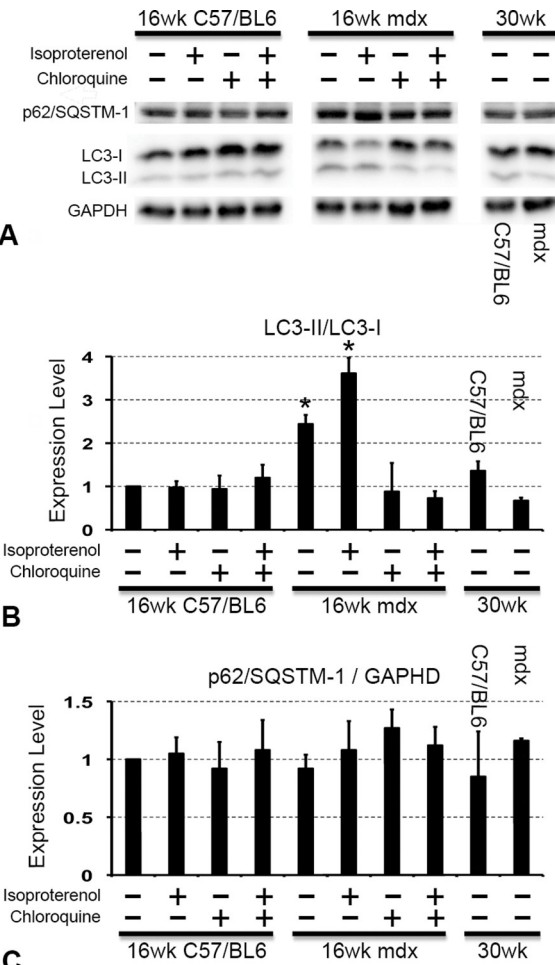

**Fig 3. LC3 and p62/SQSTM-1 expression in mdx mice cardiomyocytes. (A)** LC3 and p62/SQSTM-1 expressions in cardiomyocytes of WT and *mdx* mice. (B) Quantification of expression levels of LC3-II/LC3-I ratio. (C) Quantification of expression levels of p62/SQSTM-1/GAPDH ratio. *: *p*<0.05.

shown in Fig 3B (LC3-II/LC3-I expression ratio) and Fig 3C (p62/SQSTM-1/GAPDH expression ratio). These results were similar to the results shown in Fig 3A. The ratio of LC3-II/GAPDH was also evaluated (S1 Fig). At least the evaluation only in LC3-II, there were no significant differences between each group. These findings indicated that *mdx* mouse cardiomyocytes were more sensitive to cardiac stress than WT mouse cardiomyocytes.

## Isoproterenol-induced cardiac stress increased cardiomyocyte fibrosis significantly in only *mdx* mice and was inhibited by chloroquine

To further evaluate the cardiac stress, the fibrotic area in the left ventricle (LV) was measured after 1 month of isoproterenol-induced cardiac stress. The LV fibrotic area in 16-week *mdx* mice was remarkably larger than that of WT mice (Fig 4A (a) and (b)). Interestingly, simultaneous administration of chloroquine with isoproterenol significantly inhibited LV fibrosis in 16-week *mdx* mice (Fig 4A (c)). The calculated fibrosis area was also significantly larger in the LV of *mdx* mice. The fibrotic area of *mdx* mice was significantly reduced after the administration of chloroquine (Fig 4B). No mice treated with chloroquine exhibited side effects during the study. Furthermore, we evaluated the expression of caspase 3 in the apoptotic pathway to

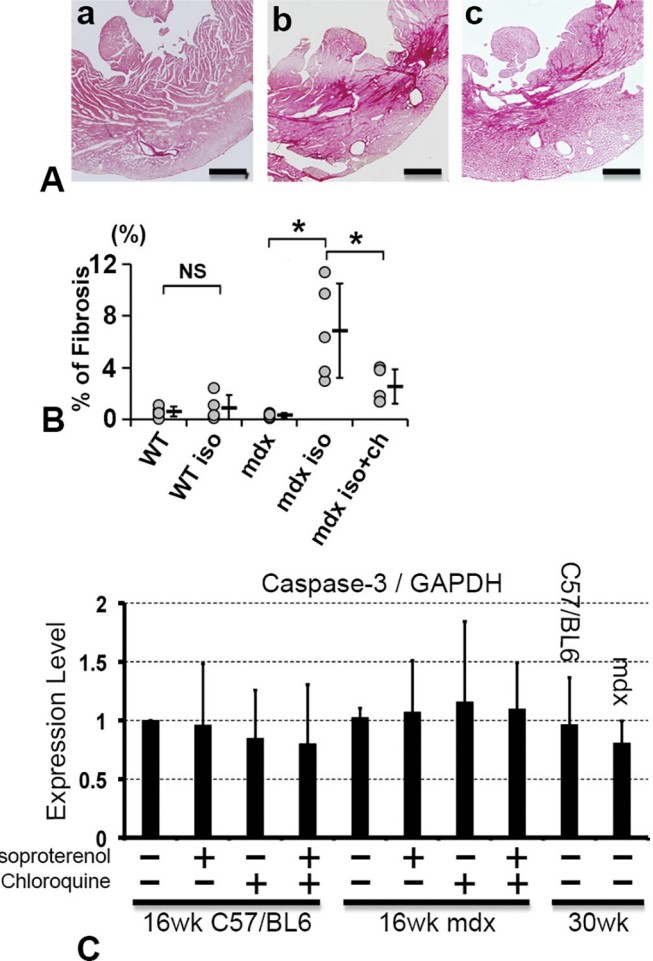

**Fig 4. Fibrosis evaluation in LV sections. (A)** LV wall fibrosis was detected by Picro-Sirius red staining. Fibrotic tissue is stained as the red area. Representative heart sections of a) an isoproterenol-treated WT mouse heart, b) an isoproterenol-treated *mdx* mouse heart, and c) an isoproterenol- and chloroquine-treated *mdx* mouse heart. Scale bar, 500 μm. **(B)** Fibrotic area in LV walls. WT, untreated WT; WT iso, isoproterenol-treated WT; *mdx*, untreated *mdx*; *mdx* iso, isoproterenol-treated *mdx* mice; *mdx* iso+ch, isoproterenol- and chloroquine-treated *mdx* mice. *: $p < 0.05$. **(C)** Quantification of caspase 3 expression level. The graph shows the ratio of caspase 3/GAPDH ratio by western blotting.

investigate another cell death. No significant changes were observed in the expression of caspase 3 in cardiomyocytes in both WT and *mdx* mice before/after isoproterenol administration and after chloroquine administration (Fig 4C). These findings indicated that cardiac stress significantly induced cardiac fibrosis in *mdx* mice, and possibly chloroquine reduced the stress-induced cardiac fibrosis.

## Chloroquine alleviated impaired cardiac contraction in *mdx* mice

To explore the effect of cardiac fibrosis and chloroquine on *mdx* mouse cardiac function, we measured the key echocardiographic markers LVEDd, LVESd, LVFS, LVEF, IVSd, and LVPWd after treatment or non-treatment with isoproterenol with or without chloroquine, as shown in Fig 1. LVEDd, LVESd, IVSd, and LVPWd were not significantly affected by isoproterenol or chloroquine in WT and *mdx* mice. (Fig 5A, 5B, 5E and 5F). However, LVFS and LVEF, markers of cardiac contraction, were significantly decreased by isoproterenol treatment

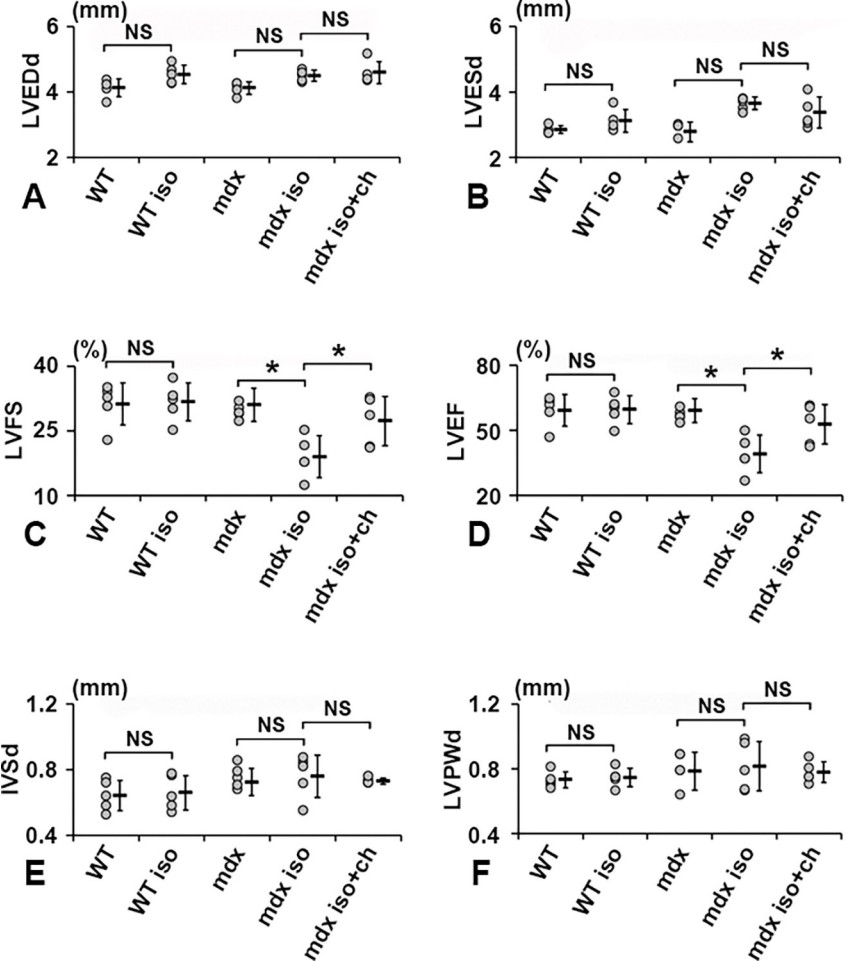

**Fig 5. LV functional measurements. (A)** LVEDd, **(B)** LVESd, **(C)** LVFS, **(D)** LVEF, **(E)** IVS, and **(F)** LVPWd were measured by echocardiography. WT, untreated WT; WT iso, isoproterenol-treated WT; *mdx*, untreated *mdx*; *mdx* iso, isoproterenol-treated *mdx* mice; *mdx* iso+ch, isoproterenol- and chloroquine-treated *mdx* mice. *: $p < 0.05$.

in *mdx* mice. Reductions of LVFS and LVEF in *mdx* mice were alleviated by simultaneous administration of chloroquine with isoproterenol (Fig 5C and 5D). Major well-characterized side effects of chloroquine, including vomiting and weight-loss, were not observed in WT or *mdx* mice. These results indicated that chloroquine protected cardiac function in isoproterenol-stressed in *mdx* mice.

## Discussion/Conclusions

In the present study, we demonstrated that cardiac stress significantly increased cardiac fibrosis in the *mdx* mouse model of DMD and was accompanied by significantly reduced cardiac contraction.

Although cardiomyopathy is a leading cause of death in DMD patients, effective therapies to prevent or slow the progression of the disease are not available [15,16]. Because the regulatory mechanisms of cardiomyopathy remain incompletely understood, current approaches to the treatment of cardiomyopathy are palliative therapies, including diuretics, angiotensin-converting enzyme inhibitors, and beta-blockers.

Recent reports have demonstrated that the mean lifespan of DMD patients is as high as 35 years of age and that more than half of DMD patients die from cardiomyopathy [12–14]. To improve the prognosis of DMD-related cardiomyopathy, mechanistic and therapeutic studies to yield more targeted approaches to treatment are necessary. In the present study, we focused on autophagic cardiomyocyte cell death using the *mdx* mouse model of DMD as well as cardiac fibrosis.

Previous reports suggested that skeletal muscle autophagy is decreased in DMD patients and *mdx* mice [19–21]. As a result of this mechanism, waste products accumulate in the cytosol, impairing skeletal muscle function [19–21]. Unexpectedly, in the present study, we identified that the number of autophagosomes was increased by cardiac stress with isoproterenol. In addition, cardiac fibrosis was increased by cardiac stress with isoproterenol. These data conflicted with prior reports of skeletal muscle autophagy in DMD. Thus, we administered the autophagy inhibitor, chloroquine, to *mdx* mice to confirm our findings. After administration of chloroquine autophagosomes dramatically decreased in cardiomyocytes of *mdx* mice under isoproterenol cardiac stress. If autophagy is assumed to be the main cause of cardiac fibrosis, chloroquine administration should further increase the number of autophagosomes. However, in this experiment, the number of autophagosomes decreased with chloroquine administration, suggesting the possibility that other effects of chloroquine, e.g. anti-inflammatory or anti-immunosuppressing effects, were thought to have suppressed cardiac fibrosis. Indeed, there is no solid evidence that autophagy suppresses cardiac fibrosis in DMD mice in our experiments, but considering that autophagosomes increase with cardiac stress, it seems possible that autophagy has an effect. In a recent study, there is a report regarding the association of myocardial autophagy with DMD [22]. Thus, in further study, the concentration of chloroquine and the administration methods of chloroquine should be considered to elucidate the mechanisms of cardiomyopathy in DMD patients.

In clinical settings, cardiac function data are the primary criteria for monitoring cardiomyopathy in DMD patients [23]. Therefore, the present study used an echocardiographic approach to evaluate cardiac function in *mdx* mice. In DMD patients, heart failure follows hypertrophic cardiomyopathy in the early stages, characterized by thinning of the ventricular wall and dilated cardiomyopathy in the late stages [24]. In accordance with clinical parameters were evaluated by echocardiography in *mdx* mice. Although left ventricular wall thinning and dilatation were not clearly observed in the present study, LVFS and LVEF, indicative of cardiac contractile function, were significantly decreased in *mdx* mice with isoproterenol cardiac stress. The lack of left ventricular wall thinning and dilatation was likely due to the relatively short duration of our studies, as one-month exposure to isoproterenol-induced cardiac stress could be too short for cardiac morphological abnormalities to develop. Nevertheless, cardiac stress impaired cardiac contractile function, which was alleviated by chloroquine. These data demonstrated that reducing cardiac fibrosis positively affected cardiac function in this context. These *in vivo* results suggest the use of chloroquine to improve cardiac function or inhibit cardiac deterioration in DMD patients. In the present study, isoproterenol was used to induce cardiac stress by increasing heart rate. Considering this effect of isoproterenol, a combination therapy of chloroquine and beta-blockers could be more effective for the treatment of cardiomyopathy in DMD patients.

The clinical application of chloroquine in DMD patients could potentially impair skeletal muscle function, as previous reports have demonstrated that skeletal muscle autophagy is suppressed in DMD [19–21]. Further studies are needed to evaluate the potential therapeutic application of chloroquine in DMD cardiomyopathy. Moreover, side effects in organs dependent on skeletal muscle function, including the respiratory system, should be carefully evaluated in the clinical application of chloroquine for the treatment of DMD-associated

cardiomyopathy. If chloroquine does not adversely affect skeletal muscle function at a certain level of blood concentration, this modality could be used to treat cardiomyopathy in DMD patients. Alternatively, modalities that specifically inhibit cardiac fibrosis could be developed.

In this study, we focused primarily on autophagy. Cardiomyocyte autophagy may occur significantly in DMD model mice, but this was not clear in detailed studies using chloroquine. At the same time, it was difficult to say that apoptosis occurred significantly in DMD model mice. However, stress-induced cardiac fibrosis clearly occurred in DMD model mice, and chloroquine administration suppressed the fibrosis, suggesting that chloroquine's anti-inflammatory effects may have contributed to the fibrosis-suppressing effect. Further studies should be conducted to evaluate the contribution of other cell death mechanisms, including necrosis and mitophagy as well as the anti-inflammatory effect of chloroquine, to cardiomyocyte dysfunction in DMD [25–27].

In conclusion, our findings demonstrated that chloroquine effects have a potential contributing mechanism to cardiomyopathy in the context of DMD and chloroquine is a potential therapeutic modality for cardiomyopathy of DMD by reducing cardiac fibrosis.

## Supporting information

**S1 Fig. Expression ratio of LC3-II/GAPDH assessed by western blotting.** No significant difference between the expression ratio of each group.
(PDF)

**S1 Raw images. Western blotting images were captured with ChemiDoc XRS Plus (BIO-RAD).** 1) LC3-I, LC3-II gel image: The expressions of 17kDa LC3-I and 13kDa LC3-II were identified by Western blotting. Cut-out images of LC3-I and LC3-II bands are shown in Fig 3. 2) p62/SQSTM-1 gel image: The expressions of 62kDa p62/SQSTM-1 were detected by Western blotting. Cut-out images of p62/SQSTM-1 bands are shown in Fig 3. 3) GAPDH gel image: 36kDa GAPDH bands were detected. The cut-out image is shown in Fig 3.
(PDF)

## Author Contributions

**Data curation:** Takuya Hirata, Shiro Baba, Katsutsugu Umeda.

**Formal analysis:** Takuya Hirata, Shiro Baba, Katsutsugu Umeda.

**Funding acquisition:** Shiro Baba, Toshio Heike.

**Investigation:** Takuya Hirata, Kentaro Akagi, Koichi Matsuda.

**Methodology:** Takuya Hirata, Shiro Baba.

**Project administration:** Shiro Baba, Souichi Adachi.

**Supervision:** Shiro Baba, Toshio Heike, Junko Takita.

**Writing – original draft:** Shiro Baba.

**Writing – review & editing:** Shiro Baba.

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
