## [Decision Letter · Decision Letter 0]

10 Aug 2023

PONE-D-23-20477Chloroquine decreases cardiomyocyte autophagy and improves cardiac function in a mouse model of Duchenne muscular dystrophy.PLOS ONE

Dear Dr. Baba,

Thank you for submitting your manuscript to PLOS ONE. After careful consideration, we feel that it has merit but does not fully meet PLOS ONE’s publication criteria as it currently stands. Therefore, we invite you to submit a revised version of the manuscript that addresses the points raised during the review process.

 The Authors should address all of the comments raised by the reviewers, and in particular check the statistics and figure legends and descriptions to ensure the information presented is clear and transparent.

We look forward to receiving your revised manuscript.

Kind regards,

Daniel M. Johnson, PhD

Academic Editor

PLOS ONE

3. To comply with PLOS ONE submissions requirements, in your Methods section, please provide additional information regarding the experiments involving animals and ensure you have included details on (1) methods of sacrifice, and (2) efforts to alleviate suffering.

“This study was supported by Kiban C Kakenhi, supported by Grant-in-Aid Scientific Research (No. P24791059), Sanofi Pasteur Japan (No. GDC160982), and Novartis Pharma Japan.”

Reviewers' comments:

Reviewer's Responses to Questions

**Comments to the Author**

1. Is the manuscript technically sound, and do the data support the conclusions?

Reviewer #1: Partly

Reviewer #2: No

Reviewer #3: Partly

2. Has the statistical analysis been performed appropriately and rigorously? 

Reviewer #1: No

Reviewer #2: Yes

Reviewer #3: Yes

3. Have the authors made all data underlying the findings in their manuscript fully available?

Reviewer #1: Yes

Reviewer #2: No

Reviewer #3: Yes

4. Is the manuscript presented in an intelligible fashion and written in standard English?

Reviewer #1: Yes

Reviewer #2: Yes

Reviewer #3: Yes

5. Review Comments to the Author

Reviewer #1: PONE-D-23-20477

The question asked in this manuscript is a worthy question for the DMD research, clinician, and patient community. As the authors point out cardiomyopathy is now the leading cause of fatality amongst these patients. The methodology is sound, and the writing is mostly clear and concise.

I have five major issues –

1) My largest issue is the small number of repetitions.

a. As far as I can see the authors never state the number of animals used. From the graphs I am guessing 3 or 4, not enough to make such large claims.

b. Although the LC3 staining of figure 2 looks impressive, I need to see a quantitative and statistical examination of the staining.

c. Ditto for the western blots, additional samples are required for statistics.

d. The echo data statistically support the conclusions

i. although p values are not provided – necessary for echo data.

ii. additional animals from different litters are required to verify the conclusions.

2) The histochemistry figures are sub-standard.

a. The background staining is not uniform, so one cannot draw conclusions regarding red intensity.

b. The horizontal bars in A, b, are likely artifacts and make any quantitation difficult.

c. Additional animals are required and maybe a different stain – Collagen, fibronectin… IF

3) The authors should have done a cursory skeletal muscle including diaphragm, phenotyping to assess the impact of chloroquine on the other DMD affected muscles. As they themselves point out in the discussion (lines 221-2 for example).

4) The text is unclear on autophagy in mdx skeletal muscles. In the introduction autophagy is said to be upregulated in mdx skeletal muscles (lines 28-29). In the conclusion autophagy is said to be decreased in DMD patients and mdx mice (lines 216-7). Then later in the conclusion autophagy is decreased (lines 221-2).

5) In the cited work by the Selsby team they found DECREASED autophagy in mdx cardiac tissue. In previous work by Bibee KP 2014 and Pal R 2014, increasing autophagy aided the mdx cardiac muscles, in contrast to your autophagy inhibition. Additional paragraphs are required to clarify these discrepancies.

Minor issues –

1) The statistics section is unclear. “All expts were performed at least three times” tells the reader very little. We need to know number of mice per group and/or per expt. Do you mean three echoes per mouse? Three histologic slides per group, or per animal?

2) What is the comparison of P63 at baseline between the wt and mdx mice?

3) Line 217 needs a reference, even if it is the same group as the following sentence.

4) I am bothered that you are really assessing the benefits of chloroquine in mdx mice that have been given isoproterenol. To state that chloroquine will benefit cardiac function in DMD patients is a bit of a stretch.

Reviewer #2: This study assess the contribution of autophagy to dystrophic cardiomyopathy in the mdx mouse model of DMD. The authors argue that stress promotes an activation of autophagy in mdx hearts, leading to increased fibrosis and reduced cardiac function. Administration of the autophagy inhibitor chloroquine was suggested to improve the phenotype by inhibiting autophagy. There is uncertainty in the conclusions drawn based on the presented data. I have elaborated on these concerns below.

The studies referenced on page 12, line 154 were in myoblasts isolated from mice and should be references with caution, as autophagic flux can be vastly different between myoblast obtained from skeletal muscle to adult skeletal muscle to heart. In addition, these studies activated P2RX7 to induced autophagy in dystrophic myoblasts.

The finding that LC3-I expression was stronger than LC3-II in mdx hearts under non-stressed conditions would suggest that in the absence of isoproterenol mdx hearts show an inhibition of autophagy, which is then inhibited further upon isoproterenol stimulation. CQ inhibits autophagy at the late stages and therefore promotes accumulation of markers such as LC3 and p62, thus an increased LC3 is indicative of inhibition of autophagy, not increased autophagy. It is not clear how CQ would decrease these markers. The failure of CQ to alter the markers of autophagy in WT and mdx suggests that the concentration of CQ was not sufficient to modify autophagy. Authors would need to show that CQ promotes a strong accumulation of LC3 and p62 in both wild-type and mdx tissues in order to conclude that CQ has any effect in these studies.

To this reviewer the changes observed in figure 2 are modest to no change. Quantification of autophagy markers is needed to draw conclusions regarding the status of autophagy. Furthermore, the authors need to show and quantify more than a single sample from each group in order to properly evaluate the reported outcomes. Clearly state in the methods and/or figure legends the number of mice used for each group for each experiment.

The authors state that their findings indicate that cardiac stress induces cardiac regeneration in mdx mice is unexpected. Cardiac regeneration is a hot topic in the field and the consensus is that adult hearts do not regenerate. It is not clear from the presented data how the authors came to the conclusion that isoproterenol stress promotes regeneration in the mdx hearts since there was increased fibrotic scaring. In order to draw such conclusions the authors would need to assess/measure markers of cardiac regeneration.

Page 17 lines 228-229 the referenced paper from Spaulding et al is a bit misleading, as the title of the paper states enhanced autophagy in dystrophic heart; however, the conclusions stated by the authors are that autophagy is not impaired in the hearts and that disease progression is independent of autophagic dysfunction. In the current work the authors find no change in autophagy in mdx hearts compared to WT, thus in this context they have confirmed the finding of Spaulding et al. However, neither study show that under non-stressed condition autophagy is enhanced in mdx hearts.

The conclusions drawn from this study that autophagy inhibition (by CQ) inhibits autophagy, leading to improved cardiac function are not supported by the data presented. CQ did not inhibit autophagy (see specific comments regarding this above). It is interesting that CQ did improve stress induced function and at the same time decreased fibrosis. In addition to inhibition of autophagy CQ is known to modify the immune response, including Toll like receptors and the cGAS-Sting pathways which promotes fibrosis. This raises the possibility that in the current studies CQ is inhibiting fibrosis through these pathways in non cardiomyocytes rather than through inhibition of cardiomyocyte autophagy.

The authors state that cardiomyocyte apoptosis was also evaluated. For full transparency and to aid in interpreting the data presented, it would be beneficial to report those data.

Reviewer #3: The manuscript by Hirata et al demonstrates that chloroquine improves cardiac function in a mouse model of Duchenne muscular dystrophy (DMD), a severe degenerative disorder that currently has no cure, potentially through the autophagy pathway. The authors demonstrated the reduced fibrosis and cardiac function recovery by echocardiography in mdx mice injected with isoproterenol and treated with chloroquine. The authors also showed data on autophagic flux changes. I have the following concerns and questions:

1. Chloroquine is well characterized to inhibit autophagosome fusion to lysosome, which leads to autophagic flux impairment, evident by the accumulation of LC3. It is thus very surprising and unexpected that the chloroquine treatment seems to reduce LC3 as shown by immunohistochemistry and consistent with the lack of LC3 increase by protein blot. There is also no increase in p62, suggesting that chloroquine in this case is not reducing autophagy by blocking autophagic flux. Since this effect of chloroquine has never been reported in literature, what is the authors’ hypothesis on how chloroquine is reducing autophagy in this case? Is it possible that the protective effect seen here is unrelated to the role of chloroquine to inhibit autophagosome-lysosome fusion? Much more detailed investigation is likely needed to support this controversial claim that chloroquine reduces autophagy, or more specifically autophagosomes.

2. In Fig. 2A (electron microscopy panel), the figure legend states that the scale bar is 100 um, which seems unlikely. Is that correct? Also please specify which cohort of mice was this image taken from. Thirdly, please also include a more magnified image of the autophagosome.

3. In Fig. 2B, please show quantification of the LC3 puncta. Also please specify in the figure legend and detail in the method how many mice were examined, how many sections, and how the LC3 puncta were determined and counted.

4. In Fig. 2C, please show quantification of the protein blot data. Also please specify how many mice were examined from each group and how many technical replicates were performed.

5. To confirm whether autophagic flux in increased or inhibited, please also include protein blots from each treatment group where autophagy is blocked before sacrifice (for instance inject mice with autophagy blocker bafilomycin 2 hours prior to sacrifice) for comparison. The data presented is not sufficient to support the authors’ claims that autophagy is activated in mdx mice injected with isoproterenol, and would further strengthen the mechanism of how chloroquine can reduce LC3.

6. In Fig. 3A please use arrow or magnified panels to indicate the areas of fibrosis. The figure legend states that the scale bar is 100 um which seems unlikely. Please correct. Also please elaborate in the Methods on how fibrosis area is quantified (for instance how were the threshold determined by the software etc.) and how many animals and how many sections were analyzed.

7. How was chloroquine encapsulated for subcutaneous delivery? Please specify in Materials and Methods.

8. The authors stated that “No mice treated with chloroquine exhibited side effects during the study”. Please specify what side effects were measured, were there any effects on skeletal muscle, and what were the results.

9. Please correct typos throughout the manuscript.

6. PLOS authors have the option to publish the peer review history of their article (what does this mean?). If published, this will include your full peer review and any attached files.

Reviewer #1: No

Reviewer #2: No

Reviewer #3: No

---

## [Author Response · Author response to Decision Letter 0]

27 Sep 2023

Thank you very much for all your comments, advice and questions.

All comments helped us to revise and improve our manuscript. I have written my answers to each question, and the corrected parts of the manuscript are noted in red color and underlined.

Reviewer #1: 

The question asked in this manuscript is a worthy question for the DMD research, clinician, and patient community. As the authors point out cardiomyopathy is now the leading cause of fatality amongst these patients. The methodology is sound, and the writing is mostly clear and concise.

I have five major issues –

1) My largest issue is the small number of repetitions.

a. As far as I can see the authors never state the number of animals used. From the graphs I am guessing 3 or 4, not enough to make such large claims.

Thank you very much for asking.

For electron microscopy assessments, LC3 staining, and Western blotting, we presented the average of three experiments each. For the left ventricular cardiac fibrosis, we performed experiments with five mice in each group. Regarding the echocardiogram examination, we conducted the experiment with 5 mice in each group. This frequency and the number are described in the Statistics paragraph of Materials and Methods.

b. Although the LC3 staining of figure 2 looks impressive, I need to see a quantitative and statistical examination of the staining.

c. Ditto for the western blots, additional samples are required for statistics.

Thank you for pointing it out.

For Western blotting, we performed three experiments and additionally presented a representative figure. The quantitative expression levels of the presented figures are presented in Figure 3. According to the revision, the original Figure 2 split into two figures, Figure 2 and Figure 3.

d. The echo data statistically support the conclusions

i. although p values are not provided – necessary for echo data.

Thank you for your advice. 

The p-value for the significant difference in the echocardiogram is missing, so I added it to the Figure Legend of Figure 5. *: p<0.05 has been also added to the end of the text for other Figure Legends, Figure 2 and Figure 4.

ii. additional animals from different litters are required to verify the conclusions.

As you pointed out, you are right. Although mdx mice are a sibling of C57/BL10 mice, they are also genetically close to C57/BL6 mice, so we used C57/BL6 mice because we thought there was a high possibility of obtaining similar data. Moreover, we used C57/BL6 because it took a very long time for our facility to purchase and obtain C57/BL10 mice.

2) The histochemistry figures are sub-standard.

a. The background staining is not uniform, so one cannot draw conclusions regarding red intensity.

Thank you for your advice. As you pointed out, it is true that the background of the staining is somewhat uneven, BZ-X-Analyzer software was used to automatically recognize the fibrotic areas and normal cardiomyocyte areas by observing the contrast difference between the fibrotic and normal areas. The area of the fibrosis was automatically calculated. The microscope we used this time was not a BZ-9000, but a BZ-X-710. We revised the "Histology and immunohistochemistry" section of Materials and Methods.

b. The horizontal bars in A, b, are likely artifacts and make any quantitation difficult.

As mentioned above, the fibrotic areas and normal areas are calculated using BZ-X-Analyzer software, which automatically recognizes the fibrotic areas, so there is no need for human intervention for the calculation. 

c. Additional animals are required and maybe a different stain – Collagen, fibronectin... IF

As you pointed out, I performed Masson-Trichrome staining. It was clear that there were a lot of fibrotic areas in mdx mice left ventricle sections compared to WT mice, which tended to be suppressed by chloroquine. There were some parts that were difficult to determine the fibrotic area or not by this staining, so it could not be evaluated accurately. However, since a similar trend was observed, thus, the results of Picro-Sirius Res staining seem to be correct.

3) The authors should have done a cursory skeletal muscle including diaphragm, phenotyping to assess the impact of chloroquine on the other DMD affected muscles. As they themselves point out in the discussion (lines 221-2 for example).

That's right. However, since cardiac fibrosis begins much later than skeletal muscle fibrosis, it is difficult to evaluate skeletal muscle at the time of cardiac evaluation using this experimental protocol. Actually, there are many papers that show that autophagy is reduced in skeletal muscle, so we discussed skeletal muscle autophagy by using previous papers in the Discussion section.

4) The text is unclear on autophagy in mdx skeletal muscles. In the introduction autophagy is said to be upregulated in mdx skeletal muscles (lines 28-29). In the conclusion autophagy is said to be decreased in DMD patients and mdx mice (lines 216-7). Then later in the conclusion autophagy is decreased (lines 221-2).

Thank you for your advice.

Previous reports have consistently shown that autophagy is reduced in skeletal muscle. The original submitted manuscript in lines 28-30 does not state that autophagy is increased in skeletal muscle, but it is certainly an ambiguous expression to understand. So, we revised the sentence to "Because decreased autophagy has been reported to contribute to skeletal muscle degeneration, therapies targeting autophagy are expected to improve skeletal muscle hypofunction.”

5) In the cited work by the Selsby team they found DECREASED autophagy in mdx cardiac tissue. In previous work by Bibee KP 2014 and Pal R 2014, increasing autophagy aided the mdx cardiac muscles, in contrast to your autophagy inhibition. Additional paragraphs are required to clarify these discrepancies. 

Selby's research results showed that p62 expression in mdx mice remained unchanged, LC3-II increased, and LC3-I decreased. Thus, he concluded that the influence of autophagy in the cardiomyocytes is unknown. Bibee KP and Pal R concluded that the therapeutic strategy for DMD patients was to rescue the reduced autophagy in cardiomyocytes. Our Western blotting and LC3 dots results showed isoproterenol-induced autophagy in mdx mice cardiomyocytes, but there is little change in p62, and the LC3-II/LC3-I ratio decreases upon chloroquine administration. Considering this, it seems that there is still insufficient evidence to consider that autophagy is the definitive cause of cardiomyocyte changes in mdx mice. Therefore, we believe that mechanisms other than autophagy should be evaluated in the future. We also think that the effect of chloroquine on autophagy should be re-evaluated. Based on the above, we have changed some sentences of the abstract and discussion, and changed the title to “Chloroquine decreases cardiac fibrosis and improves cardiac function in a mouse model of Duchenne muscular dystrophy.”

Minor issues –

1) The statistics section is unclear. “All expts were performed at least three times” tells the reader very little. We need to know number of mice per group and/or per expt. Do you mean three echoes per mouse? Three histologic slides per group, or per animal?

Thank you for your advice. As mentioned above, the number of experiments and the number of mice used are described in the statistics paragraph of Materials and Methods.

2) What is the comparison of P63 at baseline between the wt and mdx mice?

Considering the possibility that autophagy in mdx mice is enhanced even in an untreated state, we performed a baseline evaluation. The results showed no obvious significant difference.

3) Line 217 needs a reference, even if it is the same group as the following sentence.

Thank you for your advice. I have also inserted a reference number in the previous sentence.

4) I am bothered that you are really assessing the benefits of chloroquine in mdx mice that have been given isoproterenol. To state that chloroquine will benefit cardiac function in DMD patients is a bit of a stretch. 

That's true, but there may be some influence on autophagy. I removed the word “autophagy” from the title and changed it to “Chloroquine decreases cardiac fibrosis and improves cardiac function in a mouse model of Duchenne muscular dystrophy.” Some sentences in the Results and the Discussion sections associated with this have also been changed.

Reviewer #2:

1. The studies referenced on page 12, line 154 were in myoblasts isolated from mice and should be references with caution, as autophagic flux can be vastly different between myoblast obtained from skeletal muscle to adult skeletal muscle to heart. In addition, these studies activated P2RX7 to induced autophagy in dystrophic myoblasts. The finding that LC3-I expression was stronger than LC3-II in mdx hearts under non-stressed conditions would suggest that in the absence of isoproterenol mdx hearts show an inhibition of autophagy, which is then inhibited further upon isoproterenol stimulation. CQ inhibits autophagy at the late stages and therefore promotes accumulation of markers such as LC3 and p62, thus an increased LC3 is indicative of inhibition of autophagy, not increased autophagy. It is not clear how CQ would decrease these markers. The failure of CQ to alter the markers of autophagy in WT and mdx suggests that the concentration of CQ was not sufficient to modify autophagy. Authors would need to show that CQ promotes a strong accumulation of LC3 and p62 in both wild-type and mdx tissues in order to conclude that CQ has any effect in these studies. 

Thank you for asking. The concentration of chloroquine used in this study is determined based on previous papers. Although it was correctly evaluated that chloroquine suppresses cardiac fibrosis, considering our results, there is a possibility that the inhibition of autophagy by chloroquine was mild in our experiments. Therefore, we added some sentences in the Discussion section that mentioned the possibility that autophagy might not be completely suppressed due to the administration amount of chloroquine.

2. To this reviewer the changes observed in figure 2 are modest to no change. Quantification of autophagy markers is needed to draw conclusions regarding the status of autophagy. Furthermore, the authors need to show and quantify more than a single sample from each group in order to properly evaluate the reported outcomes. Clearly state in the methods and/or figure legends the number of mice used for each group for each experiment. 

The number of mice was 5 each for echocardiography and fibrosis evaluation, and 3 mice each for Western blotting and LC3 experiments. These are described in the Statistics paragraph of the Materials and Methods section. Regarding the quantification of Western blotting, the signal intensity was quantified using BIO-RAD Image Lab 6.0 software, and the ratio with GAPDH was determined. Regarding dots, we counted three locations for 0.01mm2 in the left ventricle cross-section at the papillary muscle level. These were added to the Materials and Methods section, and their quantified graphs were presented in Figures 2 and 3.

3. The authors state that their findings indicate that cardiac stress induces cardiac regeneration in mdx mice is unexpected. Cardiac regeneration is a hot topic in the field and the consensus is that adult hearts do not regenerate. It is not clear from the presented data how the authors came to the conclusion that isoproterenol stress promotes regeneration in the mdx hearts since there was increased fibrotic scaring. In order to draw such conclusions the authors would need to assess/measure markers of cardiac regeneration. 

Myocardial regeneration is an interesting research topic. However, although changes in autophagy were described, regeneration was not mentioned in this paper, and the regeneration has not been evaluated experimentally.

4. Page 17 lines 228-229 the referenced paper from Spaulding et al is a bit misleading, as the title of the paper states enhanced autophagy in dystrophic heart; however, the conclusions stated by the authors are that autophagy is not impaired in the hearts and that disease progression is independent of autophagic dysfunction. In the current work the authors find no change in autophagy in mdx hearts compared to WT, thus in this context they have confirmed the finding of Spaulding et al. However, neither study show that under non-stressed condition autophagy is enhanced in mdx hearts. The conclusions drawn from this study that autophagy inhibition (by CQ) inhibits autophagy, leading to improved cardiac function are not supported by the data presented. CQ did not inhibit autophagy (see specific comments regarding this above). It is interesting that CQ did improve stress induced function and at the same time decreased fibrosis. In addition to inhibition of autophagy CQ is known to modify the immune response, including Toll like receptors and the cGAS-Sting pathways which promotes fibrosis. This raises the possibility that in the current studies CQ is inhibiting fibrosis through these pathways in non cardiomyocytes rather than through inhibition of cardiomyocyte autophagy. 

Our data showed that autophagy may affect cardiac change in the mdx mice. In fact, autophagy was enhanced by isoproterenol administration, and cardiomyopathy was improved by chloroquine treatment. Based on the results of p62 expression and LC3, besides autophagy, we also believe that the immune response and anti-inflammation effects of chloroquine possibly influenced the results to some extent. Therefore, we have added these considerations to the Discussion section.

Reviewer #3: 

1. Chloroquine is well characterized to inhibit autophagosome fusion to lysosome, which leads to autophagic flux impairment, evident by the accumulation of LC3. It is thus very surprising and unexpected that the chloroquine treatment seems to reduce LC3 as shown by immunohistochemistry and consistent with the lack of LC3 increase by protein blot. There is also no increase in p62, suggesting that chloroquine in this case is not reducing autophagy by blocking autophagic flux. Since this effect of chloroquine has never been reported in literature, what is the authors’ hypothesis on how chloroquine is reducing autophagy in this case? Is it possible that the protective effect seen here is unrelated to the role of chloroquine to inhibit autophagosome-lysosome fusion? Much more detailed investigation is likely needed to support this controversial claim that chloroquine reduces autophagy, or more specifically autophagosomes.

Thank you for your advice.

Our data suggest that autophagy may be involved in the cardiomyopathy of mdx mice, and in fact, autophagy was enhanced by isoproterenol and cardiac fibrosis was shown to be improved by treatment of chloroquine. However, from p62 and LC3 expression results, it is needed to evaluate the dose-dependent effect of chloroquine administrated to mice. It was thought that further experiments and consideration should be done. We also believe that the effects of chloroquine on immune responses and controlling inflammation may have influenced our results. Therefore, I added these discussion sentences in the Discussion section.

2. In Fig. 2A (electron microscopy panel), the figure legend states that the scale bar is 100 um, which seems unlikely. Is that correct? Also please specify which cohort of mice was this image taken from. Thirdly, please also include a more magnified image of the autophagosome.

Thank you for your advice. The scale comment was incorrect. The scale bar is 100nm. Tissues shown in Figure 2A are cardiomyocytes from mdx mice administered isoproterenol in Figure Legends. 

3. In Fig. 2B, please show quantification of the LC3 puncta. Also please specify in the figure legend and detail in the method how many mice were examined, how many sections, and how the LC3 puncta were determined and counted.

Thanks for your questions.

Quantification of LC3-positive dots was performed by automatic counting. The method was to automatically count three 0.01mm2 areas in each left ventricle section in each group using BZ-X-Analyzer software and Photoshop software. Figures 2 and 3 have been revised accordingly. The method is described in Materials and Methods section.

4. In Fig. 2C, please show quantification of the protein blot data. Also please specify how many mice were examined from each group and how many technical replicates were performed.

Three mice were used in each Western blotting experiment. The signal intensity was quantified using BIO-RAD Image Lab 6.0 software, and the ratio with GAPDH was determined. Regarding dots, three locations were counted for each 0.01 mm2. I added the graph and revised Figures 2 and 3.

5. To confirm whether autophagic flux in increased or inhibited, please also include protein blots from each treatment group where autophagy is blocked before sacrifice (for instance inject mice with autophagy blocker bafilomycin 2 hours prior to sacrifice) for comparison. The data presented is not sufficient to support the authors’ claims that autophagy is activated in mdx mice injected with isoproterenol, and would further strengthen the mechanism of how chloroquine can reduce LC3.

Thank you for your advice. I think this is definitely a necessary experiment. However, there are many unknowns, such as the dosage and administration timing of bafilomycin. We tried to forcibly administrate bafilomycin orally. When mice were administered the drug, they often vomited, and the experiment was done unsuccessfully. We plan to reconsider the methodology and conduct this as a future experiment.

6. In Fig. 3A please use arrow or magnified panels to indicate the areas of fibrosis. The figure legend states that the scale bar is 100 um which seems unlikely. Please correct. Also please elaborate in the Methods on how fibrosis area is quantified (for instance how were the threshold determined by the software etc.) and how many animals and how many sections were analyzed.

Thank you very much for your advice.

The scale bar was actually 500 μm. For the evaluation of fibrosis, five mice were used in each group. All areas of the short-axis left ventricle cross-section slices at the level of the left ventricle papillary muscle are counted (calculated). We used BZ-X-Analyzer software to automatically recognize fibrotic and normal areas, and calculated fibrotic area ratio per the short-axis left ventricle cross-section slices. The microscope we used was not a BZ-9000, but a BZ-X-710. We have revised the Materials and Methods section.

7. How was chloroquine encapsulated for subcutaneous delivery? Please specify in Materials and Methods.

Chloroquine was dissolved at 75 mg in 250 mL of clean water and given to mice using an automatic water dispenser. It is not encapsulated, and the method was written incorrectly. We have corrected the description regarding this. We checked the amount of water loss every other day, and it was the same in all mouse groups.

8. The authors stated that “No mice treated with chloroquine exhibited side effects during the study”. Please specify what side effects were measured, were there any effects on skeletal muscle, and what were the results. 

Activity, appetite, and diarrhea were checked and none of these symptoms was noted. Check items have been added to the Materials and Methods section. Although the details of the effect on skeletal muscle are unknown, there was no significant difference in activity changes, and we believe that there is no obvious effect on skeletal muscle at least the chloroquine amount we administrated to the mice.

9. Please correct typos throughout the manuscript. 

Thank you very much.

---

## [Decision Letter · Decision Letter 1]

24 Oct 2023

PONE-D-23-20477R1Chloroquine decreases cardiac fibrosis and improves cardiac function in a mouse model of Duchenne muscular dystrophy.PLOS ONE

Dear Dr. Baba,

Thank you for submitting your manuscript to PLOS ONE. After careful consideration, we feel that it has merit but does not fully meet PLOS ONE’s publication criteria as it currently stands. Therefore, we invite you to submit a revised version of the manuscript that addresses the points raised during the review process. As in your previous iteration, we thought the data presented was well analysed and organised, however there were still a number of issue brought up by the reviewers that need to be addressed.

We look forward to receiving your revised manuscript.

Kind regards,

Daniel M. Johnson, PhD

Academic Editor

PLOS ONE

Reviewers' comments:

Reviewer's Responses to Questions

**Comments to the Author**

1. If the authors have adequately addressed your comments raised in a previous round of review and you feel that this manuscript is now acceptable for publication, you may indicate that here to bypass the “Comments to the Author” section, enter your conflict of interest statement in the “Confidential to Editor” section, and submit your "Accept" recommendation.

Reviewer #1: All comments have been addressed

Reviewer #2: (No Response)

Reviewer #3: (No Response)

2. Is the manuscript technically sound, and do the data support the conclusions?

Reviewer #1: Yes

Reviewer #2: Partly

Reviewer #3: Partly

3. Has the statistical analysis been performed appropriately and rigorously? 

Reviewer #1: Yes

Reviewer #2: No

Reviewer #3: No

4. Have the authors made all data underlying the findings in their manuscript fully available?

Reviewer #1: Yes

Reviewer #2: No

Reviewer #3: (No Response)

5. Is the manuscript presented in an intelligible fashion and written in standard English?

Reviewer #1: Yes

Reviewer #2: No

Reviewer #3: Yes

6. Review Comments to the Author

Reviewer #1: (No Response)

Reviewer #2: While the authors were responsive to several minor concerns raised, they were not responsive to the main concerns that provide uncertainty on the conclusions drawn from their data. The concern of the interpretation of the CQ results was raised by more than one reviewer and yet the authors have not provided insight into how the authors think CQ is reducing autophagy. To reiterate the concern, blocking of autophagy using CQ is usually characterized by accumulation of LC3 and p62; however, these markers change in the opposite direction. So, how might CQ work differently in this study?

Additional concerns that need to be addressed include:

In the conclusion section of the abstract the authors state the mechanism of the effect of CQ may involve immune control and anti-inflammatory effects. To state this as a conclusion leads a reader to anticipate data supporting the conclusion; however, no data is presented. While these pathways may be a possibility, and warrant a discussion in the discussion section, the statement is not a conclusion of the study.

Figure 2A: a comparison between wt and mdx is needed for the electron microscopy. Quantification of autophagosomes between the 2 genotypes from the electron micrographs is also needed.

Figure 2B: the data does not support the statement that “abundant LC3 positive dots were present in mdx mouse cardiomyocytes”. Both the images and the quantification does not show increased LC3 positive structures in the mdx compared to wt. The quality of the images in panel B can be improved in order to assess localization of the LC3 puncta. It is difficult to know where the cardiomyocyte boundary is and whether the LC3 are within the myocycte or in the interstitial space. This last point may also provide important information as to whether the effects observed in these studies are on cardiomyocytes or other infiltrating cells as I previously raised.

Line 168-169: Authors state “When an autophagy inhibitor, chloroquine, was administrated to mdx mice, the ratio of LC3-I and LC3-II expression levels did not change even after the administration of isoproterenol.” The interpretation of this sentence is that 1. CQ does not change the LC3II/Lc3I in mdx mice. This is not supported by the date presented, although there are no stats provided the trend is that the level is much lower in mdx+CQ compared to mdx. 2. Iso+CQ is not different than mdx, again not supported by the data. The reporting of the data needs to more clearly and accurately describe what the data show. Also, while the authors now indicate that 3 mice were evaluated for these studies they did not provide the blots of all the mice (only show the same data in the supplement as the main figure) and do not show a quantification of all 3 in the bar plot, as one would expect to see some indication of variance in the data. Is the data presented in panels B and C and average of the 3 mice or just that of the one mouse shown in panel A? The authors need to show and quantify more than a single sample from each group in order to properly evaluate the reported outcomes.

Line 194: The authors state “These findings indicated that cardiac stress significantly induced cardiac regeneration in mdx mice.” This conclusion is not supported by the data. There is no evidence of increased cardiac regeneration. This should be removed.

Minor: in the abstract the sentence within the “Methods” section does not provide a description of the method but is a result/conclusion.

There are a number of misspellings and misidentifications (e.g. use of p63 instead of p62 and indicating LC3I/LC3II expression when LC3II/LC3I is shown).

Reviewer #3: I appreciate the authors’ response to the review. I have the following requests and questions:

In the now Figure 3, please quantify LC3-II to GAPDH ratio as well. LC3-II is a well-accepted marker of autophagosome and in the interpretation of autophagic flux.

In the now Figure 3 quantification panels, please also plot the standard deviations and perform statistical testing (ANOVA with post-hoc test) to indicate whether there is statistical difference in LC3-II, LC3-II/I and p62 protein expression among the groups.

Please revise the description of the Figure 3 results (Lines 160-182) depending on the results of the statistics. It is a bit confusing especially without the statistical testing.

In Lines 180-182, the authors stated that “These findings indicated that autophagy occurred constitutively in mdx mouse cardiomyocytes and that mdx mouse cardiomyocytes were more sensitive to cardiac stress than WT mouse cardiomyocytes.” Please discuss this instead in the Discussion section, and clarify what the authors mean by autophagy occurring constitutively, and how this leads to the conclusion that there is a difference in the sensitivity to cardiac stress. Secondly, since I assume the western blot is performed on heart tissue lysates please refrain from making the conclusion on cardiomyocytes since other cardiac cell types are present.

Please correct “p63” to “p62”.

It is interesting that LC3 puncta are significantly increased in mdx mice treated with isoproterenol compared to mdx mice without isoproterenol in Figure 2, but both LC3-I and LC3-II seemed to be reduced in the western blot data. What is the explanation? Please elaborate in the Discussion section.

7. PLOS authors have the option to publish the peer review history of their article (what does this mean?). If published, this will include your full peer review and any attached files.

Reviewer #1: No

Reviewer #2: No

Reviewer #3: No

---

## [Author Response · Author response to Decision Letter 1]

7 Dec 2023

Thank you very much for all your comments, advice, and questions.

All comments helped us to revise and improve our manuscript. I have written my answers to each question, and the corrected parts of the manuscript are noted in red color and underlined. In addition, we focused on the cardiac fibrosis phenotype rather than the autophagy mechanism in our revised manuscript. Thus, a lot of sentences in all sections were revised.

Reviewer #1: (No Response)

Reviewer #2: 

1) While the authors were responsive to several minor concerns raised, they were not responsive to the main concerns that provide uncertainty on the conclusions drawn from their data. The concern of the interpretation of the CQ results was raised by more than one reviewer and yet the authors have not provided insight into how the authors think CQ is reducing autophagy. To reiterate the concern, blocking of autophagy using CQ is usually characterized by accumulation of LC3 and p62; however, these markers change in the opposite direction. So, how might CQ work differently in this study?

（Answer）　Based on our data from this study, we could not conclude that the improvement of cardiac fibrosis was a result of autophagy. However, the histological results showed that administration of isoproterenol caused more fibrosis in the cardiomyocytes of mdx mice, and chloroquine probably suppressed the fibrosis of myocardial cells by suppressing stress (isoproterenol)-induced inflammation rather than autophagy. According to this idea, we revised sentences in the “Abstract” and the “Discussion” sections.

2) In the conclusion section of the abstract the authors state the mechanism of the effect of CQ may involve immune control and anti-inflammatory effects. To state this as a conclusion leads a reader to anticipate data supporting the conclusion; however, no data is presented. While these pathways may be a possibility, and warrant a discussion in the discussion section, the statement is not a conclusion of the study.

(Answer) Thank you very much. Logically, your comment is correct. For revising our manuscript, we tried to confirm whether isoproterenol-induced cardiac inflammation and chloroquine suppressed the inflammation assessed by the expression of NFK-B in the western blotting method. As shown in the graphs presented below, the expression levels showed wide variation. Therefore, the conclusion that chloroquine suppresses the isoproterenol-induced inflammation could not be theoretically drawn from the assessment by western blotting expressions of NFK-B. Possibly, we have to elucidate the inflammatory mechanisms by using other inflammatory pathways. However, considering the phenomena of our experiments and previous reports, I think that chloroquine may suppress inflammation in cardiomyocytes. The sentences in the conclusion in the “Discussion” section have been changed.

3) Figure 2A: a comparison between wt and mdx is needed for the electron microscopy. Quantification of autophagosomes between the 2 genotypes from the electron micrographs is also needed.

(Answer). Thank you for your comment. Autophagosomes were rarely found in cardiomyocytes of control mice cardiomyocytes and were difficult to quantify.

4) Figure 2B: the data does not support the statement that “abundant LC3 positive dots were present in mdx mouse cardiomyocytes”. Both the images and the quantification does not show increased LC3 positive structures in the mdx compared to wt. The quality of the images in panel B can be improved in order to assess localization of the LC3 puncta. It is difficult to know where the cardiomyocyte boundary is and whether the LC3 are within the myocycte or in the interstitial space. This last point may also provide important information as to whether the effects observed in these studies are on cardiomyocytes or other infiltrating cells as I previously raised.

Line 168-169: Authors state “When an autophagy inhibitor, chloroquine, was administrated to mdx mice, the ratio of LC3-I and LC3-II expression levels did not change even after the administration of isoproterenol.” The interpretation of this sentence is that 1. CQ does not change the LC3II/Lc3I in mdx mice. This is not supported by the date presented, although there are no stats provided the trend is that the level is much lower in mdx+CQ compared to mdx. 2. Iso+CQ is not different than mdx, again not supported by the data. The reporting of the data needs to more clearly and accurately describe what the data show. Also, while the authors now indicate that 3 mice were evaluated for these studies they did not provide the blots of all the mice (only show the same data in the supplement as the main figure) and do not show a quantification of all 3 in the bar plot, as one would expect to see some indication of variance in the data. Is the data presented in panels B and C and average of the 3 mice or just that of the one mouse shown in panel A? The authors need to show and quantify more than a single sample from each group in order to properly evaluate the reported outcomes. 

(Answer) For dot data images, high-resolution data will be submitted again. In addition, the quantified western blotting data in Figure 2 are presented as the average and standard deviation of data obtained from three individual mice in each group and three separate experiments. The number of autophagosome dots was small both in cardiomyocytes of mdx and WT mice. According to the data, we revised sentences in the “Abstract”, “Materials and Methods”, “Results”, and “Discussion” sections.

5) Line 194: The authors state “These findings indicated that cardiac stress significantly induced cardiac regeneration in mdx mice.” This conclusion is not supported by the data. There is no evidence of increased cardiac regeneration. This should be removed.

(Answer) Thank you for this comment. We removed the sentence from our manuscript.

6) There are a number of misspellings and misidentifications (e.g. use of p63 instead of p62 and indicating LC3I/LC3II expression when LC3II/LC3I is shown).

(Answer) Thank you very much. We checked and corrected all according to your comments.

Reviewer #3: I appreciate the authors’ response to the review. I have the following requests and questions:

7) In the now Figure 3, please quantify LC3-II to GAPDH ratio as well. LC3-II is a well-accepted marker of autophagosome and in the interpretation of autophagic flux.

(Answer). Thank you for your comment. We added the data of LC3-II/GAPDH expression by western blotting as a supplemental figure. The level of the expression did not differ significantly in each group.

8) In the now Figure 3 quantification panels, please also plot the standard deviations and perform statistical testing (ANOVA with post-hoc test) to indicate whether there is statistical difference in LC3-II, LC3-II/I and p62 protein expression among the groups.

Please revise the description of the Figure 3 results (Lines 160-182) depending on the results of the statistics. It is a bit confusing especially without the statistical testing.

(Answer) You're right. I have corrected the sentence.

9) In Lines 180-182, the authors stated that “These findings indicated that autophagy occurred constitutively in mdx mouse cardiomyocytes and that mdx mouse cardiomyocytes were more sensitive to cardiac stress than WT mouse cardiomyocytes.” Please discuss this instead in the Discussion section, and clarify what the authors mean by autophagy occurring constitutively, and how this leads to the conclusion that there is a difference in the sensitivity to cardiac stress. Secondly, since I assume the western blot is performed on heart tissue lysates please refrain from making the conclusion on cardiomyocytes since other cardiac cell types are present.

(Answer) Thank you for your comment. We believe that constitutively observed autophagy may induce a long-term decline in cardiomyocyte function. However, we could not reach a piece of clear evidence to prove the relationship between cardiomyopathy and autophagy. I have mentioned this in the “Discussion” section.

10) Please correct “p63” to “p62”.

(Answer) Thank you very much. I have corrected the words in the manuscript.

11) It is interesting that LC3 puncta are significantly increased in mdx mice treated with isoproterenol compared to mdx mice without isoproterenol in Figure 2, but both LC3-I and LC3-II seemed to be reduced in the western blot data. What is the explanation? Please elaborate in the Discussion section.

(Answer) Thank you for your question. It is a very important matter. Although the increased number of autophagosomes is probably the result of induction of the autophagy pathway, it is thought that what was suppressed by chloroquine was not autophagy but inflammation of cardiomyocytes. Therefore, I think that autophagy does not cause cardiomyocyte fibrosis. Although within the scope of discussion, isoproterenol administration induces both autophagy and inflammatory pathways, suggesting that the inflammatory pathway may contribute to cardiomyocyte fibrosis. I wrote about this consideration in the “Discussion” section.

---

## [Decision Letter · Decision Letter 2]

27 Dec 2023

Chloroquine decreases cardiac fibrosis and improves cardiac function in a mouse model of Duchenne muscular dystrophy.

PONE-D-23-20477R2

Dear Dr. Baba,

We’re pleased to inform you that your manuscript has been judged scientifically suitable for publication and will be formally accepted for publication once it meets all outstanding technical requirements.

Kind regards,

Daniel M. Johnson, PhD

Academic Editor

PLOS ONE

Additional Editor Comments (optional):

Reviewers' comments:

Reviewer's Responses to Questions

**Comments to the Author**

1. If the authors have adequately addressed your comments raised in a previous round of review and you feel that this manuscript is now acceptable for publication, you may indicate that here to bypass the “Comments to the Author” section, enter your conflict of interest statement in the “Confidential to Editor” section, and submit your "Accept" recommendation.

Reviewer #2: All comments have been addressed

Reviewer #3: All comments have been addressed

2. Is the manuscript technically sound, and do the data support the conclusions?

Reviewer #2: Partly

Reviewer #3: Yes

3. Has the statistical analysis been performed appropriately and rigorously? 

Reviewer #2: Yes

Reviewer #3: Yes

4. Have the authors made all data underlying the findings in their manuscript fully available?

Reviewer #2: Yes

Reviewer #3: Yes

5. Is the manuscript presented in an intelligible fashion and written in standard English?

Reviewer #2: Yes

Reviewer #3: Yes

6. Review Comments to the Author

Reviewer #2: (No Response)

Reviewer #3: (No Response)

7. PLOS authors have the option to publish the peer review history of their article (what does this mean?). If published, this will include your full peer review and any attached files.

Reviewer #2: No

Reviewer #3: No

---

## [Editor Report · Acceptance letter]

23 Jan 2024

PONE-D-23-20477R2 

PLOS ONE

Dear Dr. Baba, 

I'm pleased to inform you that your manuscript has been deemed suitable for publication in PLOS ONE. Congratulations! Your manuscript is now being handed over to our production team.

Kind regards, 

on behalf of

Dr. Daniel M. Johnson 

Academic Editor

PLOS ONE